

The ST22 chronology for the Skytrain Ice Rise ice core - part 2: an age model to the last interglacial
and disturbed deep stratigraphy.
**Authors**: Robert Mulvaney[1], Eric W. Wolff[2], Mackenzie M. Grieman[2,3], Helene H. Hoffmann[2], Jack D.
Humby[1], Christoph Nehrbass-Ahles[2], Rachael H. Rhodes[2], Isobel F. Rowell[2], Frédéric Parrenin[4], Loïc
Schmidely[5], Hubertus Fischer[5], Thomas F. Stocker[5], Marcus Christl[6], Raimund Muscheler[7], Amaelle
Landais[8], Frédéric Prié[8]
1. British Antarctic Survey, Cambridge, UK
2. Department of Earth Sciences, University of Cambridge, UK
3. Reed College, Portland, Oregon, USA
4. Université Grenoble Alpes, CNRS, IRD, Grenoble INP, IGE, 38000 Grenoble, France
5. Climate and Environmental Physics, Physics Institute, and Oeschger Centre for Climate Change
Research, University of Bern, Switzerland
6. Laboratory for Ion Beam Physics, ETH Zurich, 8093 Zurich, Switzerland
7. Department of Geology, Quaternary Sciences, Lund University, Sölvegatan 12, SE-22362 Lund,
Sweden
8. Laboratoire des Sciences du Climat et de l'Environnement, LSCE/IPSL, CEA-CNRS-UVSQ, Université
Paris-Saclay, Gif-sur-Yvette, France
Correspondence to: Eric Wolff (ew428@cam.ac.uk)

## 1. Abstract

We present an age model for the 651 m deep Skytrain Ice Rise ice core. The top 2000 years have
previously been dated using age markers interpolated through annual layer counting. Below this, we
align the Skytrain core to the AICC2012 age model using tie points in the ice and air phase, and apply
the Paleochrono program to obtain the best fit to the tie points and glaciological constraints. In the
gas phase, ties are made using methane and, in critical sections, $\delta^{18}O_{air}$; in the ice phase ties are
through $^{10}Be$ across the Laschamps Event, and through ice chemistry related to long-range dust
transport and deposition. This strategy provides a good outcome to about 108 ka (~605 m). Beyond
that there are signs of flow disturbance, with a section of ice probably repeated. Nonetheless values
of $CH_4$ and $\delta^{18}O_{air}$ confirm that part of the last interglacial (LIG), from about 117-126 ka (617-628 m),
is present and in chronological order. Below this there are clear signs of stratigraphic disturbance,
with rapid oscillation of values in both the ice and gas phase at the base of the LIG section. Based on



methane values, the warmest part of the LIG and the coldest part of the penultimate glacial are
missing from our record. Ice below 631 m appears to be of age >150 ka.

2.      Introduction

There is currently intense interest in the role of the Antarctic Ice Sheet, and the West Antarctic Ice
Sheet (WAIS) in particular, in future sea level rise (DeConto et al., 2021; Fox-Kemper et al., 2021).
While modern studies of the behaviour of the WAIS are essential, studies aimed at assessing the past
stability of the WAIS and its response to past climate change are required to constrain the operation
of proposed feedbacks (such as the Marine Ice Cliff Instability mechanism) (Gilford et al., 2020).  The
last interglacial (LIG, Marine Isotope Stage (MIS) 5e, ~130-110 ka before present (bp) where present
is defined as 1950) has been considered of particular interest because estimates of sea level during
that period compared to the present (Dutton et al., 2015; Dyer et al., 2021) appear to require some
contribution from retreat of the Antarctic Ice Sheet.  In order to assess the sensitivity of the WAIS
and its surrounds to climate change, it is also of interest to understand how the climate and the ice
in the WAIS region responded to the coolings and warmings of the last glacial period and the
warming into the Holocene.
While there are a number of Antarctic ice core records extending through at least one climate cycle
and into the LIG from East Antarctica (e.g. Crotti et al., 2021; EPICA Community Members, 2004;
Grootes et al., 2001; Kawamura et al., 2007), long records from West Antarctica are scarce. The
WAIS Divide ice core (Fig. 1) provides an excellent and well-resolved record of the last 68 kyr (Buizert
et al., 2015) but does not extend further back in time. The only other long core in the interior of the
WAIS is the 2191 m long Byrd core, for which the oldest ages presented are 90 ka (Ahn and Brook,
2008). On the periphery of the WAIS, the Siple Dome core reached the bed at 1004 m, but again data
have only been presented as far back as 90 ka (Brook et al., 2005; Saltzman et al., 2006).  At
Roosevelt Island, situated within the Ross Ice Shelf, the ice could not yet be dated beyond 83 ka (Lee
et al., 2020). Old ice might be available at the bottom of the Berkner Island (Mulvaney et al., 2007)
and Fletcher Promontory (Mulvaney et al., 2014) cores, but there is no published age scale for these
cores so far.



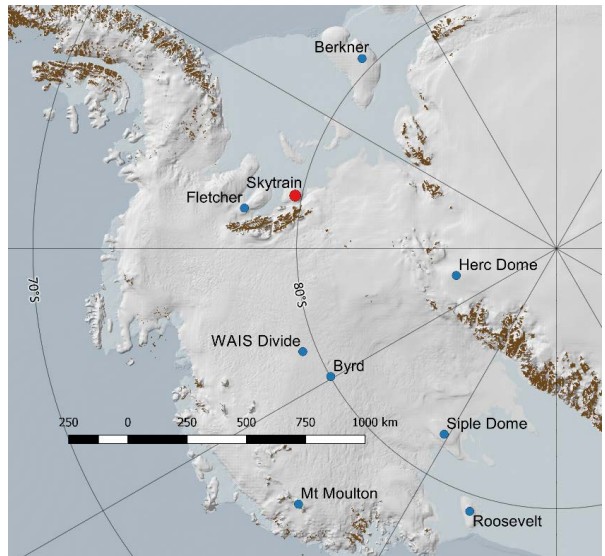


Figure 1. Map showing ice core sites in West Antarctica that are mentioned in text. Map generated
using QGIS with the Quantarctica mapping environment (Matsuoka et al., 2021).

The only record that seems to unequivocally reach the LIG in West Antarctica to date is that from a
horizontal ice trench in the blue ice area at Mount Moulton (Korotkikh et al., 2011). This appears to
reach 135 ka, although the nature of the record makes it hard to assess its continuity.  It is therefore
a priority to find sites in the WAIS vicinity where a record extending to the LIG can be retrieved and
fully analysed. One potential candidate site, near the boundary between the East and West Antarctic
Ice Sheets, would be Hercules Dome (Jacobel et al., 2005), and drilling is expected there in the next
few years. In this paper we present an age scale for an ice core drilled at Skytrain Ice Rise, at the
boundary of the WAIS and the Ronne Ice Shelf.



The core at Skytrain Ice Rise was drilled to the bed at 651 m depth in 2018-19 (Mulvaney et al.,
2021). Skytrain Ice Rise (Fig. 2) is an independent ice rise (i.e., with its own flow regime) with a
circular shape and a diameter of ~80 km. It sits at an altitude of 784 m, has a 10 m temperature
(representing mean annual temperature today) of -25.9°C, and a basal temperature of -14.9°C. It
represents an attractive target because it's isotopic and chemical content should be sensitive to
changes in the extent and altitude of the WAIS, and also to the extent of the adjacent Ronne Ice
Shelf. It is situated on a bed that is above sea level, but surrounded almost entirely by ice shelf
(including Constellation and Hercules Inlets, see Fig. 2) that has a sea bed depth of at least 1000 m.
On the WAIS side, it is protected by the Ellsworth Mountains. This combination ensures that Skytrain
Ice Rise will almost certainly have remained as a separate ice dome, and would never have been
overridden by inland ice, whatever the size of the WAIS.

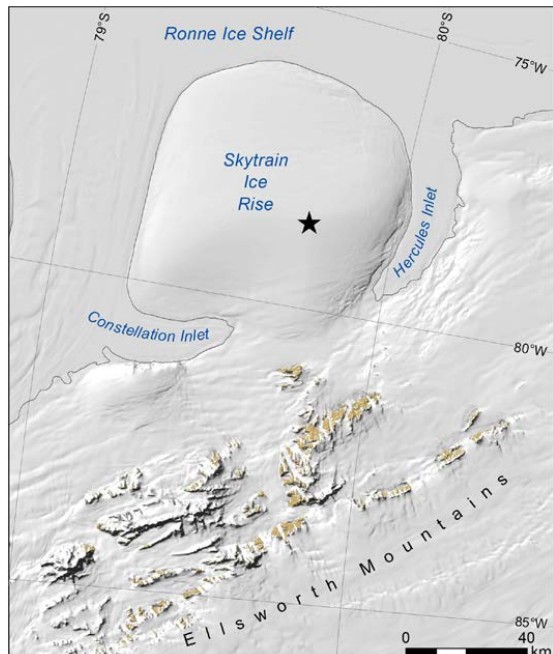


Figure 2. Skytrain Ice Rise. The drill site is marked with a star. Figure reproduced from (Mulvaney et
al., 2021), CC BY 4.0
Radar data collected previously (J. Kingslake, pers. comm.) showed good layering almost to the bed
(Mulvaney et al., 2021), with a pronounced Raymond Arch. The drill site was chosen based on the
radar layers to give old ice as far from the bed as possible.



In a companion paper to this one (Hoffmann et al., 2022) we have used a variety of age markers,
interpolated through counting of annual layers in chemistry, to derive an age scale for the last 2000
years (~200 m). In this paper we use a range of evidence to derive an age model for the rest of the
core. In particular, we demonstrate that the core contains an intact record of the last glacial period
and extends into the LIG. We also discuss the possible age of more disturbed ice found in the
deepest twenty metres of the core.

3.  Overall dating strategy
The strategy, as with other recent dating papers (Epifanio et al., 2020), is to tie the Skytrain Ice Rise
core to a well-established age model. Since we expected our core to run well beyond the age of the
WAIS Divide core, we have chosen to give our final derived ages as those of the AICC2012 age model
(Bazin et al., 2013; Veres et al., 2013), which was developed for the EPICA ice cores but includes
synchronized age scales for some of the major East Antarctic Ice Sheet deep ice cores (Talos Dome,
Vostok) and which is synchronized to the Greenland NGRIP ice core in the upper 60 kyr. However,
we recognise that the WD2014 age model (Buizert et al., 2015; Sigl et al., 2016), developed for the
WAIS Divide ice core) is more accurate in absolute age over the last 68 kyr, and that methane data
are available at a much higher resolution in cores that have been tied to it. For that reason, in some
cases we initially matched our core to WD2014 and then used a simple translation table to tie it to
AICC2012. For convenience, our depth-age table in the supplement provides both WD2014 and
AICC2012 ages for the last 68 kyr. This is based on volcanic synchronisations (Buizert et al., 2018; Sigl
et al., 2022) for the age of the ice.
In order to construct the age alignment, and estimate uncertainty, we use the Paleochrono program
which is a development of the Icechrono program (Parrenin et al., 2015). We include a number of
stratigraphic alignments to AICC2012, based on the data in the companion paper for the uppermost
2000 years, and using $CH_4$, $\delta^{18}O_{air}$, $^{10}Be$, and ice chemistry markers in deeper ice. Paleochrono was
started with a prior for the accumulation rate (based on a simple relationship with water isotope
ratios), air lock-in depth (prior set at a constant 58 m (Hoffmann et al., 2022)) and a simple ice
thinning function. Paleochrono minimises a cost function that measures the misfit of the model with
respect to the prior and the observations (tie points).
3.1.  Flow disturbance
In the deeper part of the ice core, between 628-635 m, we observe some discontinuities, with rapid
and simultaneous changes in water isotopes and methane at the same depth. These will be



discussed in more detail later, but they represent likely depths of flow disturbance or folding, as has
been observed in other ice core records, including those of the LIG in Greenland (Chappellaz et al.,
1997; NEEM Community Members, 2013; Yau et al., 2016). We also deduce that some disturbance
may exist in a region between about 605 and 615 m depth. From 600 m downwards we therefore
carefully examine individual data points (using paired values of $CH_4$ and $\delta^{18}O_{atm}$ matched against
reference data) to reconstruct discrete ages for particular depths. This allows us to assess which
sections are in order with well-constrained ages, and which are disturbed in the deeper ice. We then
use Paleochrono to derive a continuous age model to 630 m, making manual adjustments to the
final age scale to avoid assigning spurious ages to data in the disturbed section.

4.   Data available

In this section we describe the collection of the data used to make ties to other cores, both in the
gas phase (air bubbles) and in the ice phase.

137       4.1.  Continuous methane

Methane measurements are a particularly powerful way of aligning the gas ages of different ice
cores because they exhibit large (from tens to 200 ppb) and abrupt changes of concentration across
millennial-scale Dansgaard-Oeschger events that recur throughout the last glacial period (e.g.
Epifanio et al., 2020). Using high-resolution continuous analysis it has also been shown that
centennial and faster variability down to below 10 ppb amplitude is well-reproduced between cores
(Lee et al., 2020; Mitchell et al., 2013; Rhodes et al., 2017). As methane is well-mixed in the Antarctic
troposphere, not just the pattern but the absolute values should match with reference datasets
within uncertainty. Our main dataset, the continuous one from CFA, is good at showing the high-
resolution variability, but has a large and unknown uncertainty in absolute values. We therefore
supplement it with some discrete analyses (section 4.2) that constrain the concentration tightly at
key sections of ice.
We measured methane ($CH_4$) continuously during the continuous flow analysis (CFA) campaign
(Grieman et al., 2021). Briefly, the core was melted at a mean rate of 3.2 cm min$^{-1}$ and the air was
separated from residual water flow using a 3M Liqui-Cel MM-0.5x1 Series membrane contactor.  The
dried air was then directed to a Picarro G2301 CRDS for $CH_4$ analysis. While the methane Picarro
calibration could not be checked against external certified standards, comparison of our data



produced by CFA with analysis of discrete samples analysed in Bern (section 4.2), as well as
comparison of our CFA data with reference data across the Holocene and glacial, suggests that the
CFA methane reproduces the variability in methane at centennial scales. However, the absolute
values are offset (mainly low) by an amount that varied by a few percentage points over the
campaign but the offset was typically below 10%. This offset arises partly from dissolution of a small
percentage of gas into the meltwater stream, as has been observed previously using CFA to measure
methane (Rhodes et al., 2015). Continuous analyses started at 244 m depth and continued in all
sections where the ice was of suitable quality to 649.4 m. A short section from 144.0-161.3 m was
also analysed continuously for methane with an improved measurement setup which is discussed in
the companion paper (Hoffmann et al., 2022).
Two significant issues affected the measurements. Firstly, a section of data between 534 and 545 m
was affected by a leak of lab air at the membrane contactor. The absolute values in this section of
ice are therefore substantially higher than palaeoatmosphere, but the pattern of variability can still
partly be used for wiggle-matching after correction using discrete analyses (next section).
A second issue is that there were increasing numbers of breaks and cracks in the ice with depth,
particularly below 450 m. Badly cracked sections were removed before the ice was placed on the
melter and breaks across the core were smoothed with a cleaned file to ensure that the contact
between ice sections was as close as possible. With these precautions, such occurrences do not
affect the ice phase chemical measurements and most do not affect $CH_4$ either. Nonetheless some of
the remaining cracks and transitions between different bags provide an opportunity for the ingress
of lab air as the ice melts, leading to spikes in methane concentration. Major short peaks and
troughs were identified using the "ginput" MATLAB function, and removed from the dataset.  Above
500 m ~25 spikes that were at least a factor 2 higher or lower than the mean of the dataset were
removed.  Below 500 m, the data became much noisier and ~100 deviations from the dataset were
manually removed. Even after removal of the obvious spike artefacts the data remain more noisy
than the data that are unaffected by such artefacts, suggesting that positive artefacts arising from
inclusion of modern air remain in the dataset. This makes it trickier to clearly align data with a
reference dataset in the deeper ice. The dataset below 244 m is shown in Fig 3.





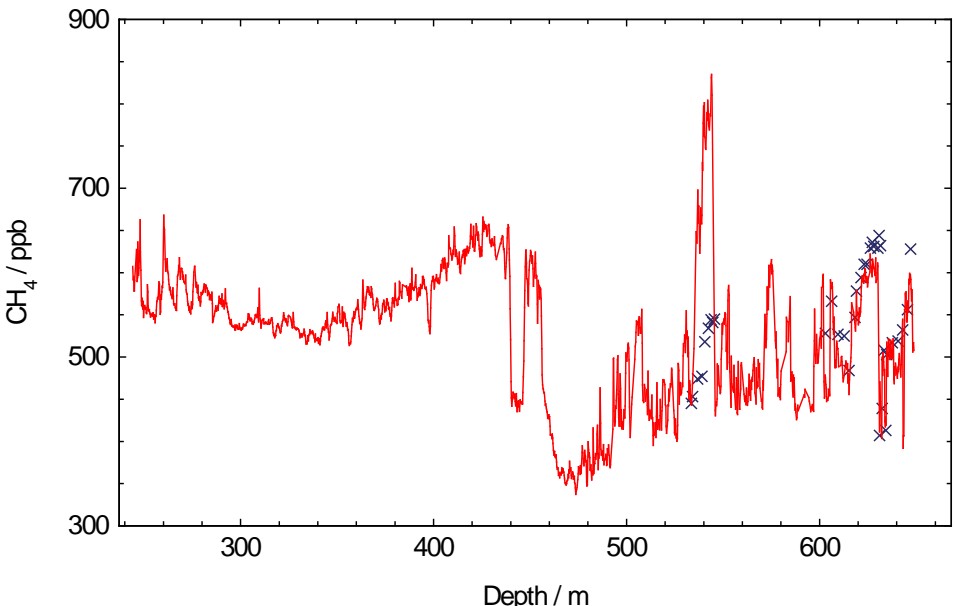


Figure 3. Continuous (CFA) methane (red), and data from discrete measurements (black crosses),
after removal of occasional methane spikes as discussed in the text. The discrete data confirm that
the continuous data between 534 and 545 m are offset, and confirm that the uncalibrated values for
the remaining continuous data are reasonable.

187       4.2.  Discrete methane

To validate and control that the absolute levels of our continuous $CH_4$ record are consistent within
uncertainty with the absolute values in reference data, we obtained some well-calibrated discrete
measurements (Fig. 3), particularly in the deep ice and in the section impacted by the air leak
(section 3.1). Ten discrete samples were therefore measured at the University of Bern between 533-
546 m, and a further 25 samples between 600 and 650 m depth. Details of the method have been
published elsewhere (Schmidely et al., 2021). Concentrations ranged between 413 and 644 ppb,
with an estimated precision (1 sigma) of 7 ppb (Table S1). Note that the discrete data presented here
have been corrected by -18 ppb (Schmidely et al., 2021) to align them with previously published $CH_4$
records. These offsets are potentially due to different remnant solubility of $CH_4$ in meltwater using
different melt extraction methods in different labs. Taking the uncertainty of the correction into
account, the total uncertainty is estimated at 12 ppb (Schmidely et al., 2021), while that of the
reference data is estimated at 10 ppb (Loulergue et al., 2008). Combining these uncertainties
suggests that when comparing absolute values of methane (discrete data) with reference datasets
we should allow an uncertainty of 16 ppb (much higher offsets are possible for the data derived by





CFA, and there we mainly look for similar patterns to those in the reference data).  The discrete data
measured in Bern are displayed along with the continuous data in Fig. 3. A number of discrete
measurements were also made between 84 and 144 m at Oregon State University which are
described in the companion paper (Hoffmann et al., 2022).
### 4.3. $\delta^{18}O$ of $O_2$ ($\delta^{18}O_{atm}$)
The isotopic ratio of oxygen in air provides a good additional constraint because it is well-mixed
globally, and varies in line with precession, providing opportunities for aligning measurements with
calculated orbital targets as well as with measurements from other ice cores (Extier et al., 2018).
$CH_4$ and $\delta^{18}O_{atm}$ have previously been used powerfully in tandem to untangle disturbed ice
chronologies in the LIG (Chappellaz et al., 1997; Yau et al., 2016).
In this work, 27 samples were analysed for $\delta^{18}O_{atm}$ at the Laboratoire des Sciences du Climat et de
l'Environnement (LSCE). Two samples were in the depth range 160-170 m, and 5 were between 435
and 471 m. The remaining samples were in the depth range 602-635 m. Data were corrected for firn
fractionation and gas loss  (Extier et al., 2018) and are shown in Table S1.  Uncertainty on each value
is estimated at +/- 0.03 ‰. Combining this with the similar uncertainty in data points in the
reference dataset suggests that we should allow an uncertainty of 0.04‰ when comparing our data
with the reference.
### 4.4.  $^{10}Be$ across the Laschamps Event
The flux/concentration of $^{10}Be$ in ice shows a pattern related to variations in the magnetic field of
the Sun and, on longer timescales, Earth.  The pattern of these variations can be matched between
ice cores, and with $^{14}C$ variations in other archives such as tree rings, in order to synchronise records
(e.g. Adolphi and Muscheler, 2016). A particularly clear and prominent pattern is seen across the
Laschamps Event, a weakening of Earth's magnetic field that occurred around 41 ka bp (e.g. Raisbeck
et al., 2017). Because this section of ice is in the last glacial period, its synchronisation in the ice
phase should allow for a particularly useful and unambiguous estimate of the offset between ice age
and gas age (Δage) in the glacial period.
Seventy samples from between 509 and 520 m depth were spiked with a known amount of $^{9}Be$,
processed in Lund and analysed for $^{10}Be$ by Accelerator Mass Spectrometry at ETH Zurich. Measured
$^{10}Be/^{9}Be$ ratios were normalized to the ETH Zurich in-house standards S2007N and S2010N with
nominal $^{10}Be/^{9}Be$ ratios of $28.1 \times 10^{-12}$ and $3.3 \times 10^{-12}$ (Christl et al., 2013). Data and associated
uncertainties are presented in Table S2.
### 4.5.  Aluminium (Al) and non sea salt magnesium (nssMg)





When synchronising ice cores from different sites, it is important to use only parameters for which
there is a sound reason to assume that both cores share synchronous variability. This is the case, for
example, with volcanic eruption spikes, with [10]Be and with well-mixed atmospheric gases, such as
methane. It is not safe to make such an assumption for water isotopes, which are site-dependent
because climatic changes may vary asynchronously in different parts of Antarctica. While methane
synchronisation (see above) and a relatively small Δage compared to inland sites (due to the higher
accumulation rate) allows us to make a reasonable estimate of the ice age along our core, it would
be advantageous to have further ties in the ice phase.  It has been argued previously that variations
in the components of terrestrial dust (such as Ca) can be assumed to be synchronous across
Antarctica (Baggenstos et al., 2018; Mulvaney et al., 2000). This is because their concentrations are
strongly controlled by events at a common source in South America and in a common part of the
transport pathway towards Antarctica, with only a minor part of the variability likely to be
dependent on the final stages of transport to each ice core site.
The main component used for such synchronisation to date has been non-sea-salt (nss) Ca,
calculated using marine and terrestrial ratios of Ca and Na (e.g. Röthlisberger et al., 2002)). However,
after an initial attempt we observed that while nssCa at Skytrain Ice Rise shows a good coherence
with that of other sites (EDC, EDML) until a depth of about 500 m (30 ka bp), it diverges below that.
Other terrestrial markers such as Al and nssMg (calculated as Mg-0.12*Na and both measured by
ICP-MS during the CFA campaign (Grieman et al., 2021)) do not mirror the Skytrain nssCa signal, and
do appear to follow nssCa at other East Antarctic sites (see section 6.3).  It appears that an additional
source of Ca-rich material, not seen in other Antarctic cores and presumably due to local sources, is
present at this site in the earlier part of the last glacial.  The reasons for this will be explored
elsewhere. However, the solution for us is to use the terrestrial markers that appear free from this
extra source, but that are coherent with nssCa records at other sites. The limits of detection of Al
and Mg are 3.3 ppb and 1.3 ppb, respectively. We concentrate on alignments from nssMg because a
majority of Al values in the Holocene and marine isotope stage 5 fall below the detection limit; in the
glacial the Al values support our conclusions with nssMg.
5.   Reference datasets
Since the basis for our age model is tying variations in our data to variations in well-dated ice cores,
in this section we describe the reference datasets used.

264       5.1.  Gas phase: Methane and $\delta^{18}O$ of $O_2$ ($\delta^{18}O_{atm}$)



In order to use the more detailed variability that can be traced during the Holocene, we compared
our methane data to the high resolution Roosevelt Island methane record between 2-7 ka bp.
Between 7-68 ka we used the WAIS Divide record (Buizert et al., 2015; Rhodes et al., 2017). Between
68 and 156 ka, we used the southern hemisphere methane spline generated from the EDC ice core
(Köhler et al., 2017). To investigate possible matches with older ice we used the EDC data itself
(Loulergue et al., 2008). As previously explained, the Roosevelt and WAIS Divide data are on the
WD2014 age scale, but we eventually used a conversion table (based on Buizert et al., 2018) to place
all matches onto a common AICC2012 age scale.
A composite EDC-Vostok record of $\delta^{18}O_{atm}$ (Extier et al., 2018) was used for comparison to Skytrain
ice core $\delta^{18}O_{atm}$.

275        5.2.  Ice phase: $^{10}$Be across the Laschamps Event and terrestrial marker elements

The clear pattern of the $^{10}$Be record across the Laschamps Event has been shown to be closely
replicated at several sites in Greenland and Antarctica (Raisbeck et al., 2017). For the
synchronisation, we used the normalised stack that was recently created based on 3 Greenland and
3 Antarctic records (Adolphi et al., 2018).
As the reference dataset for terrestrial deposition we used the nssCa record from EDML (Fischer et
al., 2007), because of its greater proximity to Skytrain in the Atlantic sector of Antarctica, with
further validation using the record from EDC (Wolff et al., 2010).
6.    Tie points to 100 ka

284        6.1.  Methane

First, we note that the discrete methane data (Fig. 3) confirm that the methane concentrations in
the section from 534-545 m are much too high. In this section of ice we therefore use the values
from the discrete data to match with reference data.
In Table S3, we list the methane tie points that we used in this section.  The very clear match
between our record and the reference data is ideally seen in the past 15 kyr (460 m) where there are
few spikes in the methane record due to air ingress into cracks (Fig. 4). However, the pattern of
Dansgaard-Oeschger events remains clear right down to 100 ka, and is shown in Fig. 5, along with
the tie points used. We note that the comparisons in Fig. 4 suggest that the Skytrain data might be
up to 10% too low in concentration (but with a variable offset along the core) compared to the
reference data; this results from the dissolution of gas in the meltstream (as discussed in section 4.1)
and the difficulty of accurately calibrating data from the continuous melter due to the absence of an
external certified standard. In Fig. 5 we show the full methane record on the eventual age scale,





compared to reference data. It is clear that some spikes due to air ingress across cracks remain in the
dataset beyond about 60 ka, but the pattern for matching is still apparent to at least 100 ka. The
section beyond 100 ka will be discussed in section 7.

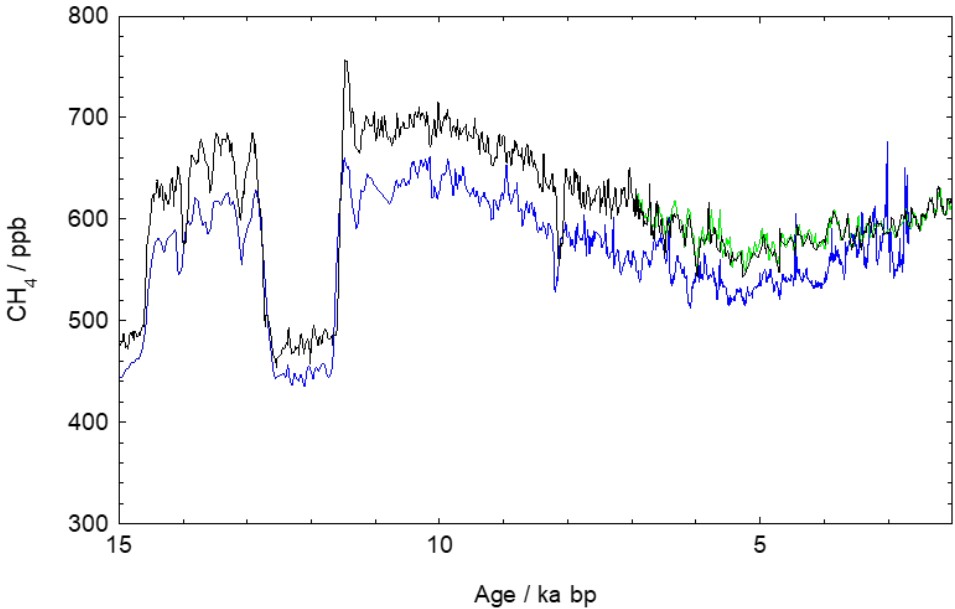


Figure 4. Methane matching over the last 15 kyr. Methane from Skytrain Ice Rise (blue) on its age
scale after synchronisation, along with methane from Roosevelt Island (green) (Lee et al., 2020), and
WAIS Divide (black) (Buizert et al., 2015; Mitchell et al., 2013). Ages shown here are WD2014. The
concentration offset between the Skytrain and other data is probably caused by partial dissolution in
the meltstream for Skytrain as discussed in the text.

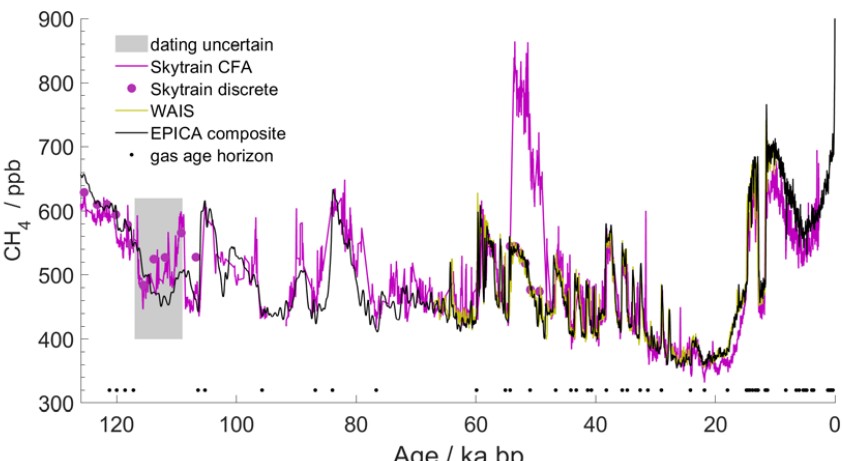


Figure 5. Methane from Skytrain Ice Rise on the ST22 age scale, along with reference data. Skytrain is
shown in purple (continuous is a line, discrete data as dots). In black is a spline of Antarctic data
(Köhler et al., 2017). WAIS Divide is shown in yellow (Buizert et al., 2015; Mitchell et al., 2013). Ages
shown here are AICC2012. Gas age tie points are shown along the bottom of the figure. The grey
shaded area represents the ice (605-617m) with unreliable ages due to flow disturbance (see section

313 9).

6.2. $^{10}$Be across the Laschamp Event
In Fig. 6 we show the Skytrain $^{10}$Be concentration from 509-520 m, aligned with the reference
dataset. The common shape across the wider event as well as the presence of individual peaks and
troughs is clear. We chose 5 tie points in the range 39.9-42.0 ka bp (Table S4).

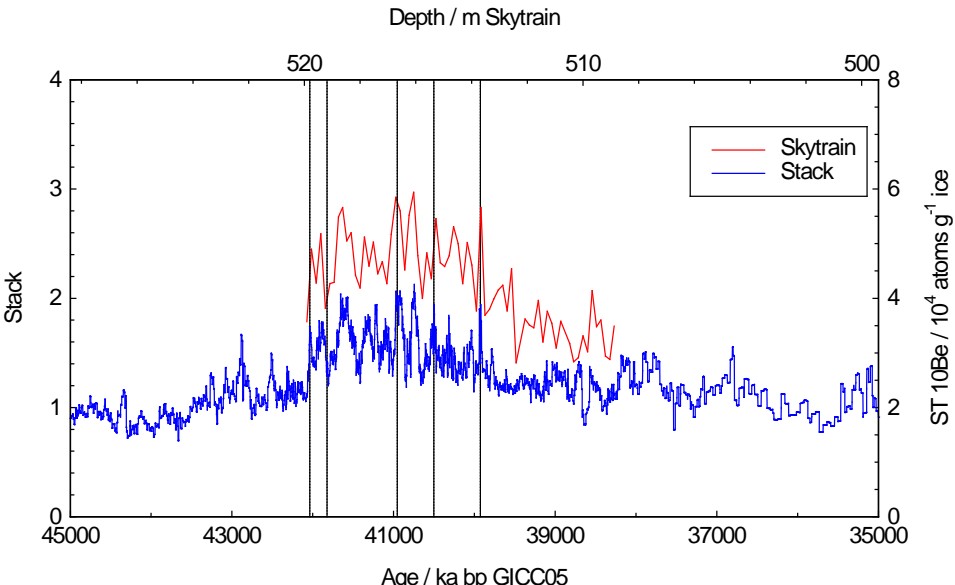


Figure 6. $^{10}$Be concentration in the Skytrain (ST) ice core (red) compared to the normalised stack of

ice core radionuclide data (Adolphi et al., 2018).  Two samples with obvious low outlier

concentrations in the ST record have not been plotted. Vertical lines show the tie points used in this

study.

### 6.3.  nssMg compared to Ca at EDML

Skytrain nssMg was compared to nssCa from EDML (Fischer et al., 2007) (Fig. 7). The two records

show strong similarities, as does Skytrain Al (not shown) where it exceeds the detection limit;

comparison with EDC nssCa (Wolff et al., 2010) shows a comparably good match.  We chose a few

obvious tie points (Table S4) concentrating on regions with clear variability and trying to fill the gaps

where fewer ice tie points existed.  We discuss the ice below 100 ka in section 7.




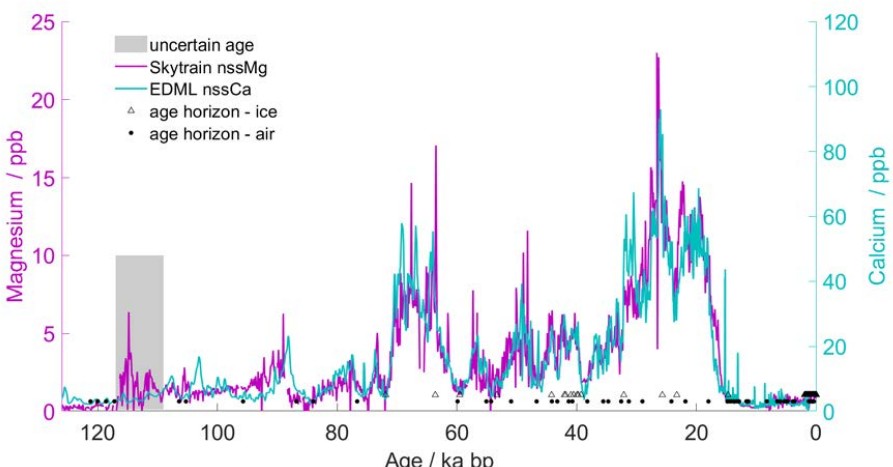


Figure 7. nssMg at Skytrain shown on its age scale after synchronisation (purple). nssCa from EDML

(cyan) (Fischer et al., 2007). Tie points used in this paper are shown (circles are gas age, triangles are
ice age ties). The grey shaded area represents the ice (605-617m) with unreliable ages due to flow
disturbance (see section 9).

7.   Dating the ice older than 100 ka
Below about 600 m (100 ka), methane continues to show a pattern similar to that of the reference
record, with a peak between 600-603 m (Fig. 3) that seems to correspond to the methane peak
associated with Greenland interstadial (GI) 24 at 102-107 ka (Baumgartner et al., 2014; Capron et al.,
2010). However below this, between 605-608 m, there is a further methane peak that appears
anomalous: its concentrations are too high to match the reference data at GI 25. Whereas methane
peaks typically have a sharp jump in concentration at their old (deeper) side, this peak has a sharp
drop at its shallower side. From 616 to 622 m, methane rises in a stepped fashion similar to the
increase seen in the reference record on the young side of the LIG between 114 and 123 ka, before
plateauing (~625-629 m) at concentrations typical of the last interglacial (as confirmed by the
discrete measurements made in Bern, with several concentrations between 630 and 644 ppb).
However there are no values (in either the continuous or discrete data) that reach those (going
above 700 ppb) that are seen in the reference data in the early last interglacial peak between 127
and 129 ka.  Additionally, methane experiences a rapid alternation of values (two values > 600 ppb
surrounding a value of 400 ppb within a metre) at 631 m (the base of the values that appear to be
interglacial). This coincides (in depth) with a rapid alternation in water isotope ratios (not shown
here). Finally there are also very few values below 400 ppb that would correspond to the low values
seen in the reference data during the penultimate glacial maximum between about 140-145 ka.





These observations suggest that the ice is in good chronological order to 107 ka and probably from
about 117-126 ka, but that there might be a flow disturbance between 107 and 117 ka, and a
definite disturbance and discontinuity at the base of the last interglacial ice with some thousands of
years potentially missing from our record. Later we speculate on the reasons for this. For now it
causes us to be concerned about the integrity of the record above this depth (ie the LIG to 126 ka). It
suggests that the use of simple pattern matching of methane and nssMg in the LIG ice might risk a
false assignment, and so instead we seek a more definite quantitative match.
7.1.  $CH_4$ and $\delta^{18}O_{atm}$
Flow disturbances affecting LIG ice have been seen previously, though until now this has been
observed mainly in Greenland. To confirm the age of ice with difficult stratigraphy, and even to re-
order disordered layers, previous authors have used a combination of methane and $\delta^{18}O_{atm}$
(Chappellaz et al., 1997; NEEM Community Members, 2013; Yau et al., 2016). Provided data are
sufficiently precise, the two-dimensional field of these parameters can define an age for a given
layer that is close to unique within the plausible range. In Fig. 8 we show the reference data for $CH_4$
and $\delta^{18}O_{atm}$.

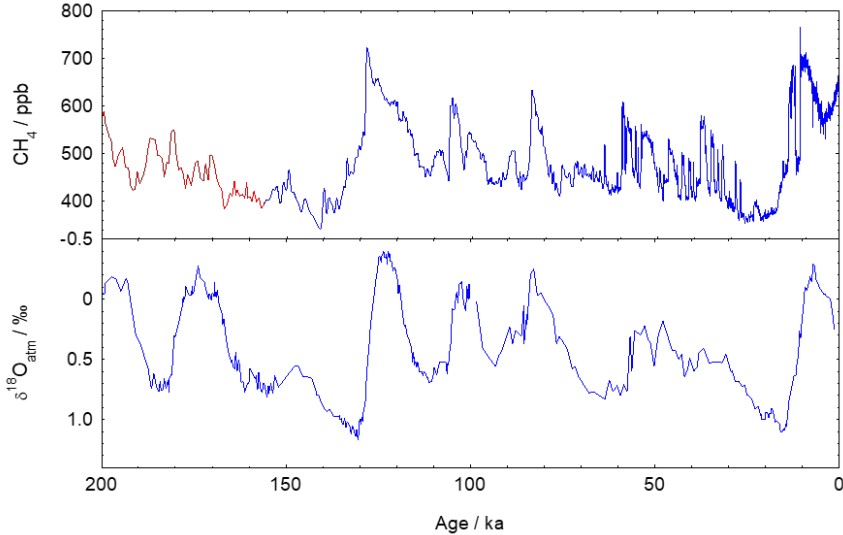


Figure 8. Reference data for $CH_4$, to 156 ka in blue (Köhler et al., 2017), beyond 156 ka in red
(Loulergue et al., 2008) and $\delta^{18}O_{atm}$ (Extier et al., 2018) over the last 200 ka. Data are all on the
AICC2012 age model.



By plotting the two-dimensional distribution of values (Fig. 9) one can see how the data clearly
differentiate samples of different ages – this is particularly true in the section from about 120-150 ka
(section that goes clockwise in increasing age from mid-blue to green). While the $\delta^{18}O_{atm}$ data were
used mainly in combination with $CH_4$ to assess the ages of ice around the LIG, $\delta^{18}O_{atm}$ was also
measured in two Skytrain ice core samples from the Holocene and two from the last glacial
maximum: these were not used to construct the age scale but the values were entirely consistent
with the modelled ages. Three samples were also measured between 435 and 456 m. These three
values of $\delta^{18}O_{atm}$, along with the less precise $CH_4$ data obtained from the continuous measurements,
were used to assign ages (Table S1) more precisely between 11 and 15 ka in a section in which
$\delta^{18}O_{atm}$ is increasing rapidly with age (Fig. 8).

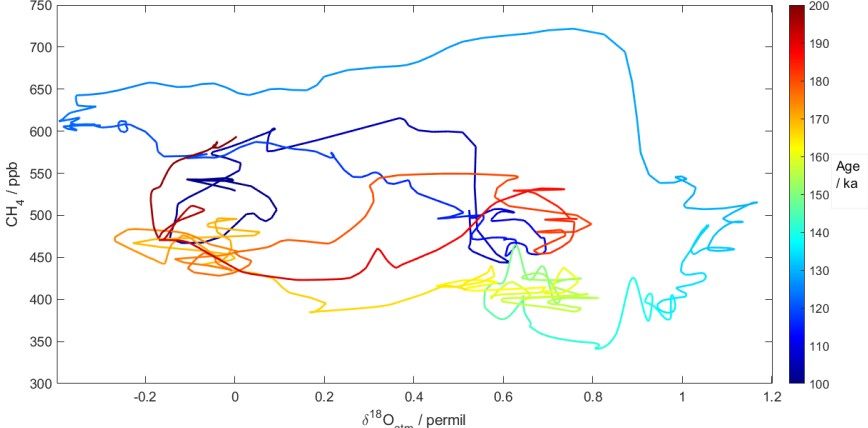


Figure 9. Cross plot of $CH_4$ (Köhler et al., 2017; Loulergue et al., 2008) and $\delta^{18}O_{atm}$ (Extier et al., 2018)
reference data for the period 100-200 ka. The colourbar indicates the age of the sample.

Twenty Skytrain ice core samples were analysed for $\delta^{18}O_{atm}$ between 600 and 635 m depth, covering
the period that the discussion above would lead us to expect is older than 100 ka. In all but two
cases discrete methane measurements were made (in Bern) on an adjacent sample (a few cm away
from the $\delta^{18}O_{atm}$ sample).
We now examine the data at depths for which we have both $\delta^{18}O_{atm}$ and methane measurements.
We start with the data from 603-618 m (Fig. 10). The data point at 603.1 m can be assigned an age of
~106 ka, as we had already deduced above from the shape and amplitude of the methane peak
alone. While the point at 606.4 m matches best with ~118 ka, the 3 data points deeper than that



(609-618 m) are only compatible with younger ages, between 106 and 117 ka.  We cannot untangle
this section but there is apparently some degree of disturbance at least between 605 and 615 m.
Turning now to Fig. 11, the data from 615.3 to 627.3 m plot in chronological sequence with respect
to the reference data between about 110-126 ka. Most of these points are not consistent with
$\delta^{18}O_{atm}$ and methane values at any other ages in the range, 60-180 ka.  Crucially the two datapoints
at 623.2 and 624.7 m with very negative $\delta^{18}O_{atm}$ and CH$_4$>600 ppb are not compatible with any other
age in the past 200 kyr other than the LIG at around 122 ka, and a short period in the Holocene at 7
ka.  These datapoints are also incompatible with any mixtures of ice from other depths. Because the
data point at 615.3 m is compatible with a range of ages, we choose a conservative range of depths
from 617 m (just above the clear match at 618.3 m) to 628 m where  we are very confident that we
have a sequence of ice from the last interglacial, covering the period 126 ka to 117 ka. Although it
lies within the uncertainty of the values at 627.3 m, the data point at 630.3 m (shown in red) is also
only consistent with the last interglacial, but does not show the expected increase in age with depth,
and could show a reversal in age. As this is already in the section that appears disturbed in methane
and $\delta^{18}O_{ice}$, we consider this data point and the ice around it as subject to disturbance.

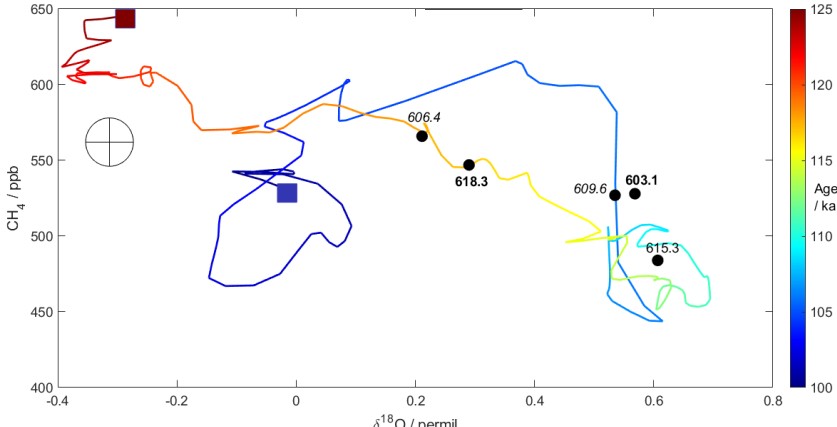


Figure 10. Cross plot of CH$_4$ (Köhler et al., 2017) and $\delta^{18}O_{atm}$ (Extier et al., 2018) reference data for
the period 100-125 ka, along with Skytrain Ice Rise data from 603-618 m depth (black  dots). The
combined uncertainty (used to decide whether a match between the Skytrain and reference data is
acceptable) is shown by the grey ellipse/cross. The start (125 ka) and end (100 ka) of the reference
curve are marked by red and blue squares. Skytrain data points are marked with depths; the ones
we later judge as being in disturbed ice are marked with italics, while the ones we consider well-
dated are in bold.



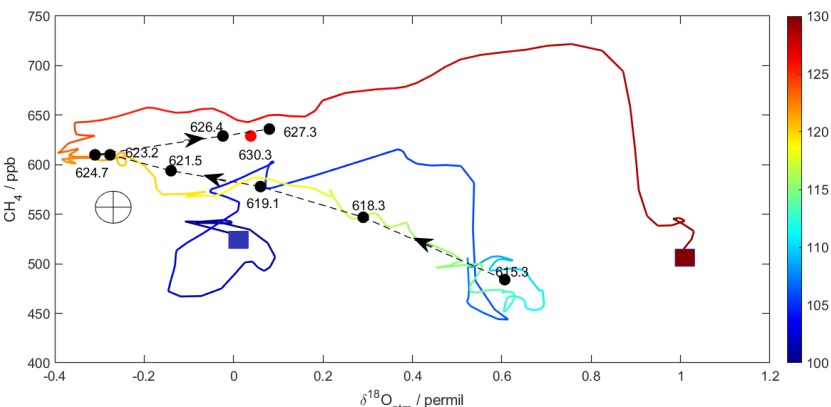


Figure 11. Cross plot of CH$_4$ (Köhler et al., 2017) and δ$^{18}$O$_{atm}$ (Extier et al., 2018) reference data for
the period, 100-130 ka, along with Skytrain Ice Rise data from 615-628 m depth (black dots joined by
dashed line with arrows pointing in order of increasing depth) and 630.3 m (red dot). The combined
uncertainty (used to decide whether a match between the Skytrain and reference data is acceptable)
is shown by the grey ellipse/cross. The start (130 ka) and end (100 ka) of the reference curve are
marked by red and blue squares.

Finally, we examine the data from 630 m to 635 m (Fig. 12). The point at 631.6, sitting close to
clearly disturbed ice with rapidly changing values of CH$_4$ and δ$^{18}$O$_{ice}$, has values not seen in the
reference data, and is probably a mixture of interglacial and glacial ice. The other data have values
consistent with ages that would occur in the middle of MIS6 (140-180 ka), or alternatively could
originate from ice that is much older (from an earlier glacial cycle). Because there are a number of
age solutions within the uncertainty of the measurements we do not attempt to assign ages to these
data points.

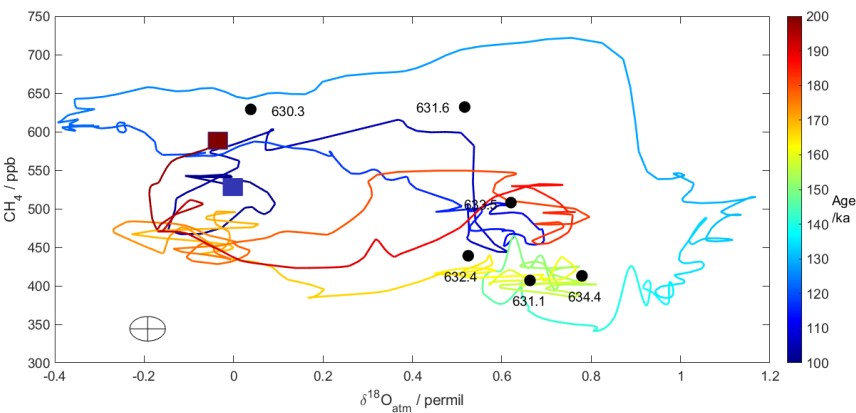






Figure 12. Cross plot of $CH_4$ (Köhler et al., 2017; Loulergue et al., 2008) and $\delta^{18}O_{atm}$ (Extier et al.,
2018) reference data for the period 100-200 ka. The colourbar indicates the age of the sample. Also
shown are the Skytrain data from 630 m downwards (black dots). The combined uncertainty (used to
decide whether a match between the Skytrain and reference data is acceptable) is shown by the
grey ellipse/cross. The start (200 ka) and end (100 ka) of the reference curve are marked by red and
blue squares.

7.2.  Stratigraphy around the LIG
Combining the observation that no ice has methane values that fit in the age ranges 127-129 ka or
~140 ka, and the positive identification of ice with unique combinations of $CH_4$ and $\delta^{18}O_{atm}$, we
conclude the following:
a) there is probably a flow disturbance at the top of the last interglacial section, with ice from ~106-
117 ka repeated;
b) despite this, there is a continuous section of ice from 617-628 m that represents the time period
from 117-126 ka in good order;
c) there is strongly disturbed ice at the base of the LIG section, with the ice below it most likely
representing much older ice from MIS6 or beyond.

8.  Application of Paleochrono
The Paleochrono model was run using the prior constraints discussed in section 3 and the tie points
described in sections 6 (and shown in Table S3 and S4). For the section deeper than 600 m we have
assigned tie points based on $CH_4$ and $\delta^{18}O_{atm}$ that anchor 603 m at 106 ka, and ties for each
$CH_4/\delta^{18}O_{atm}$ pair between 617 and 628 m (117-126 ka). We then assigned a much older age to 632 m
just to allow continuity of the age scale to the bed. No other tie points were applied below 628 m
(126 ka), and the ice ages below that were ignored. Between the tie points at 603 and 618 m,
Paleochrono assigns ages but because we know that there is disturbance and likely repeated ice, we
cannot trust all of them. As a compromise, in our age scale we report the ages as far as 605 m (108.7
ka) and from 617 m (~117 ka) but do not show any ages for 605-617 m.  The age model is reported
with both ice age and gas age, along with uncertainties derived from the model. Fig. 13 shows the
depth-age relationship (continuous line) from the model. A depth-age lookup table is supplied in the
supplement. Methane and nssMg data are shown on the derived age model to 126 ka in Figs. 5 and
7. We have placed a grey bar on data in the disturbed section (605-617 m) where ages cannot be
considered reliable.

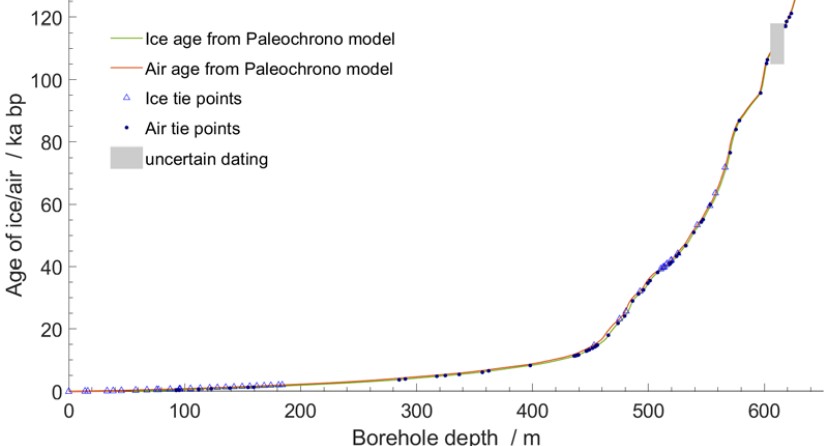

Figure 13. Age against depth for the Skytrain Ice Rise ice core. Ice and air age are shown, along with the tie points we applied. The section with unreliable ages (605-617 m) is greyed out.

In the supplement we present the deposition rate (Fig S1) and thinning function (Fig. S2) derived from the model. No dramatic deviations are seen, indicating that the derived age model is physically reasonable. However given the flow disturbances beyond 605 m the derived deposition rate and thinning may be unreliable from 605 m to the bed.

To further assess the age assignments around the LIG, in Fig. 14 we show the values of discrete measurements of $CH_4$ and $\delta^{18}O_{atm}$ with the ages from Paleochrono for the sections of ice we consider less disturbed. It can be seen that both the values and sequence for both parameters are consistent, and generally match the reference data within uncertainty between 117 and 126 ka. Although Palechrono separated them in order to maintain continuity, the data points (at 626.4 and 627.3 m), showing as slightly displaced from the reference curves at 125 and 127 ka in Fig. 14, were originally both assigned tie point ages of ~126 ka, which would also place them on the reference curves.





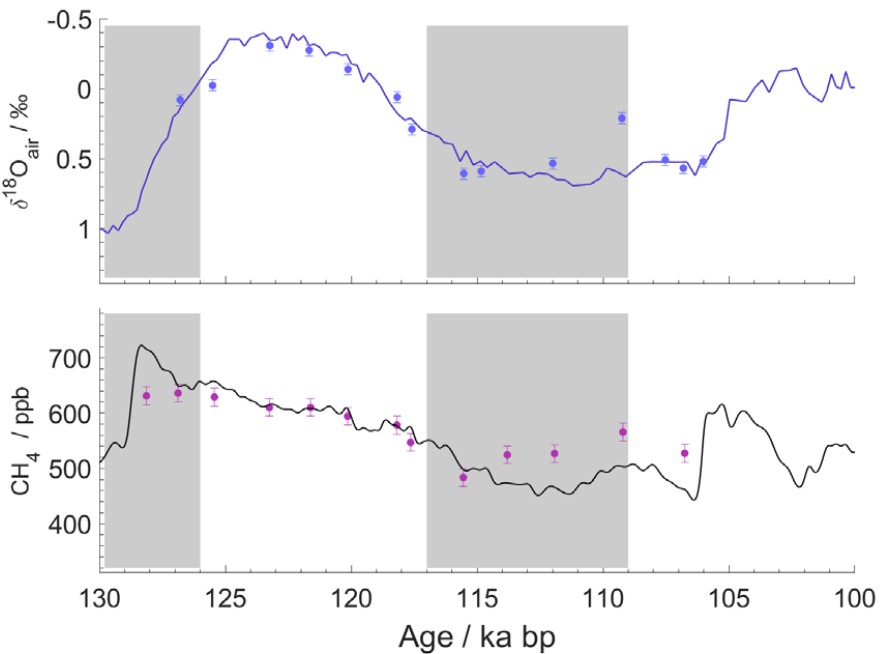

482

Figure 14: Reference data for CH$_4$ and $\delta^{18}$O$_{atm}$ between 100 and 130 ka (as in Fig. 8), along with

discrete measurements (symbols) for the Skytrain Ice Rise ice core. Sections with unreliable ages

(605-617 m and >628 m) are greyed out. The error bars are the combined uncertainty (at 1 sigma) of

the Skytrain and reference data.

9. Disturbed ice around the LIG

It is evident that there is ice disturbance both at the top, and particularly at the base, of the LIG.

Such disturbances have been observed in previous LIG ice, though until now only documented in

Greenland ice (Grootes et al., 1993; NEEM Community Members, 2013). Such discontinuities have

been hypothesised to result from the contrast between ice layers with very different rheological

properties, due to changes in impurity content and grain size (LIG versus Penultimate Glacial

Maximum (PGM) and LIG versus late MIS 5) (NEEM Community Members, 2013). We expect smaller

contrasts in properties in Antarctica compared to Greenland. However, a tendency to become

disturbed and folded might be exacerbated at Skytrain Ice Rise by the existence of a rather large

Raymond arch (Mulvaney et al., 2021), a dynamic feature seen in the radar profiles, extending right

to the bed (the internal layering (Mulvaney et al., 2021) shows upwarping of order 50 m within



around 1 km horizontal distance only 100 m above the bed). Although we expect Skytrain Ice rise to
have remained a separate flow centre, it is likely that the position of the dome was different during
the LGM when the Ronne Ice Shelf would have been grounded and provided greater constraint to
the north and east; this could also have led to disturbance around the LIG ice which would already
have been deep in the ice column at that time.
We consider here possible alternative causes for the hiatus, with ice from 127-129 ka missing from
our sequence, and probably ice from 129 to at least 140 ka also unrepresented.
a) The first possibility is that there was no snow accumulation during this period. This is
considered extremely unlikely. The section from 127-129 ka at other Antarctic sites shows
high temperatures and inferred high accumulation rates.
b) A second possibility is that the ice from inland overrode Skytrain Ice rise causing some layers
to be removed completely. However, the Ellsworth Mountains provide a high and rather
solid barrier against such flow. There is also no sign of ice anywhere in the core with the
much more negative water isotopic contents one would expect from ice originating at much
higher altitude inland.
c) Some ice sheet models have inferred a possible loss of ice from parts of WAIS during the LIG
(DeConto and Pollard, 2016). This hypothesis raises the possibility that ice was completely
lost from Skytrain Ice Rise in the warmest part of the LIG. However, the existence of more
than 20 m of ice that appears to derive from MIS6 or older suggests that ice was not
completely removed from Skytrain Ice rise. In addition if some ice was lost by melting, while
older ice was retained, we would expect to see bubble-free ice (caused by refreezing after
melting). This is not observed anywhere in the core: normal values of total air content and
methane concentrations are seen at all depths.
We therefore conclude that the only plausible explanation for our observations is flow
disturbance due to contrasting rheology.

10. Conclusion
We have constructed an age model, which we call ST22, for the Skytrain Ice Rise ice core. This age
model is based mainly on tie points to previous Antarctic ice cores, using a range of analyses. The
age-depth relationship is well-behaved until at least 100 ka. There appears to be flow disturbance at
the top of the LIG section, but the core contains ice from the last interglacial (117 to 126 ka) in good



stratigraphic order. It is however missing the earliest part of the LIG, and the coldest part of the
PGM, apparently also due to flow disturbance affecting ice layers with contrasting rheologies.
Because the missing ice appears to have been affected by flow disturbances, we surmise that
another core at a suitably chosen location on Skytrain Ice rise might be capable of retrieving ice from
the missing sections. This is the first time that flow disturbances around the LIG have been clearly
documented for Antarctica, as they have been several times for Greenland. These disturbances raise
the possibility that such disturbances might also have affected other records of the LIG (Korotkikh et
al., 2011). One obvious conclusion from our data is that the ice sheet was certainly present at
Skytrain Ice Rise during the LIG.
**Data availability**
The continuous methane and nssMg used in this paper (and shown in Figs 5 and 7) have been
submitted to Pangaea. The discrete $CH_4$, $\delta^{18}O_{atm}$ and $^{10}Be$ data used in this paper are attached as
supplementary data (Tables S1 and S2). The air and ice tie points used in Paleochrono are attached
as supplementary data (Tables S3 and S4). All reference data used in this paper are already
published and available online.  The final derived age model ST22 is attached as supplementary table
S5, and has been submitted to Pangaea.
**Author contributions**
The first two authors contributed equally to this paper. The paper was written by RMul and EW with
contributions mainly from HH, MG and RR. The ice core was drilled and sectioned by EW, RMul, CN-
A, MG, IR. The CFA analysis was performed by HH, MG, JH, RMul, RR and IR.  Discrete methane
analyses were provided by LS, HF and TS; $\delta^{18}O_{atm}$ data were provided by FP and AL; $^{10}Be$ data were
provided by MC and RMus. RMul ran Paleochrono with advice from FP. All authors contributed to
improving the final paper.
**Competing interests**
The authors declare that they have no competing interests.
**Acknowledgments**
The authors thank Shaun Miller, Charlie Durman, Amy King, Emily Ludlow, Liz Thomas and Victoria
Alcock for help with cutting, processing and analysing the ice core. This project has received funding
from the European Research Council under the Horizon 2020 research and innovation programme
(grant agreement No 742224, WACSWAIN). TS and LS acknowledge funding from the Swiss National
Science Foundation (#172745 and #2000492), and all authors from the University of Bern gratefully



acknowledge the long-term support of ice core science by the Swiss National Science Foundation.
This material reflects only the authors' views and the Commission is not liable for any use that may
be made of the information contained therein. EW and HH have also been funded for part of this
work through a Royal Society Professorship. The development of Paleochrono was funded by two
CNRS/INSU/LEFE projects called "IceChrono" and "CO2Role".

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
