# Peer review of "The ST22 chronology for the Skytrain Ice Rise ice core part 2: an age model to the last interglacial"

_Climate of the Past, 2022_

## Author Comment (AC1)

Reviewer 1

Review in normal type, response in *italic*:

Mulvaney et al. present a timescale for the Skytrain Ice Rise ice core which covers all ice older than 2 ka. The timescale to ~106 ka is continuous and based on methane matches using the continuous methane record supplemented by discreet measurements. They then identify an area of ice flow disturbance. Using discrete measurements of d18O of O2 and methane from the same depths, they confirm a flow disturbance, date a section below the disturbance to ~106-126ka and suggest a continuous climate record during this interval. Below this, the ice is again disturbed and likely from the penultimate glacial period. Interestingly, both the onset of the Last Interglacial and the Penultimate Glacial Maximum are missing, which the authors suggest is due to flow disturbances caused by contrasts in ice fabric.

 The paper is important, well written and should be published with minor revisions. This paper will be foundational for what I image are future high impact papers on the climate and ice sheet interpretation for which a timescale is necessary. The authors have made a wide range of measurements and performed a thorough analysis of the core. The timescale is well developed with a lot of care taken to explain the approach. I have some recommendations for making this even clearer, but I appreciate the effort the authors have taken. I have only two area to suggest significant changes. The first is the timescale from about about 75 ka to 109 ka. The second is the uncertainty of the timescale.

I also want to specific note that I appreciate that the authors have made the data publicly available.

*We thank the reviewer for this generous summary.*

First, one initial thought on the introduction. The authors lead the introduction with a statement about the "intense interest" in the stability of WAIS and the need for paleoclimate records to constrain potential ice sheet changes. Later they write "old ice might be available … Berkner Island and Fletcher Promonotory, but there is no published age scale for these cores so far". The lead author of this paper was the project leader for both of those ice cores. I hope that this work on Skytrain inspires those cores to be revisited and published.

 *Yes, indeed. The lead author's colleagues have made the same point to him!*

**Discussion of timescale from ~75ka to 109ka, particularly 95ka to 109 ka**

The timescale for this period seems more uncertain than the authors imply and could use more description. I realize much of the emphasis of this paper is on the LIG, but I think it is important to discuss where the end of the continuous climate record is reached.

The methane matches are not particularly robust given how different the shapes are, like the wider width of the peak centered on ~82 ka. The subpeak at 80 ka is also distinctly different. The nssMg does seem to help in this period.

 The nssMg match is less robust in the period 95 ka to 109ka. At 95 ka, the methane rise at Skytrain looks substantially larger than the EPICA composite. Since there are folds in this area, I think more explanation should be given about why the methane features at ~95 ka and ~106ka are not a repeat of younger events.

I think a new section which discusses this interval would be help. With more description, I might agree that the timescale is continuous to 109 ka.

*The reviewer is correct to point out that the methane matches in the region 80-100 ka are not as robust as those in shallower ice. This arises mainly from the poor quality of the methane record because the ice was so cracked, meaning that occasional undetected air ingress was unavoidable, and that data are missing over some of the detailed methane features.*

*Taking the specific areas highlighted by the reviewer: the width of the peak at 82 ka arises from our decision to align the records on the lowest points at ~77 ka and 87 ka, with only one alignment point in between. We acknowledge that we placed too small an uncertainty on the depths of these alignments. We have tested a new age model experiment where we have raised the uncertainties of these 3 alignments to 1000 years, and (see response to next section) raised the uncertainty on the thinning function. This pulls in the width of the peak on the old side and makes the ST data consistent with the reference data within our new (greater) uncertainties.*

*The subpeak at 80 ka is hard to diagnose, because it consists of about 40 cm of ice, with 73 cm of missing data just above it. It's therefore difficult to decide whether it belongs where you see it and represents a too high concentration due likely to air entering cracks on the melter, or whether it belongs to the left where it would match the reference subpeak at 81 ka but have a rather larger dip than expected. Again by increasing the uncertainty we can accommodate either scenario.*

*We have implicitly assumed that the spikes at 90 ka that take the ST record higher than the reference are undetected fliers that should be ignored (they represent only a few cm of the record).*

*We recognise that the match in this section using methane alone is not as convincing as we would wish. However we would argue that the good match of nssMg with reference nssCa (in both values and age) between especially 75-90 ka support our alignment – remember that we imposed no ice tie points in this region of the core, so this is completely independent support.*

*For the section to 100 ka, we will add some text to section 6 to acknowledge the nature of our match to 100 ka. Our proposed additional text is: "The match between Skytrain and reference methane between 80 and 100 ka is less secure than it is in shallower ice, because ice with high concentration outliers and/or missing data is common as a result of extensive cracking. This makes it hard to match absolute values of methane, and forces us to rely on the pattern with depth. Nonetheless, the methane ties we have made result in a good match in this part of the core between nssMg and reference nssCa (Fig. 7 and section 6.3), supporting our choices."*

*Between 95 and 109 ka, we agree that we are relying almost entirely on methane, although the low concentrations of nssMg rule out most other sections of glacial ice. We do have the single del18O_air/discrete methane pair at 106 ka, which has values that don't occur again after 106 ka until 57 ka. Proposed additional text: "We have no reason to doubt that the ice is in good order until 605 m, but we acknowledge that the section we date as 95-107 ka (Figs. 5, 7) relies on the pattern of methane and on a single $CH_4/ \delta^{18}O_{atm}$ datapoint. This point, dated at 106 ka, firmly defines the lower end of this section, with values that do not occur again as a pair until 57 ka."*

*In summary, in a revised version we have increased the uncertainties in our tie points and as a result we have larger uncertainties in the final age model, which can now accommodate the comparison seen in Fig. 5. We will also add some sentences acknowledging the difficulties in this section in more detail.*

**Uncertainty**

The uncertainty is not discussed substantially in the manuscript. The uncertainties are given as part of the paleochrono output in the supplement, but I think these are likely too low. They assume a continuous climate record with well behaved ice-dynamics and confident tie points. I think this is likely only justified to about 75 ka. Given the meter to decameter scale folding, there is almost certainly smaller scale folding as well. Thus, the assumption of continuously increasing age even in sections that are primarily intact is not a given (hopefully the stratigraphy is being looked at for a future paper because it could be fascinating). And the confidence in the tie points seems to drop. The 82 ka methane peak is a good example. The uncertainty is given as <300 years but it looks like there is a 1000 year offset between the midpoints of the rise. I suggest two things:

- the uncertainty be manually adjusted to be greater for the older portions of the core. This can be a qualitative uncertainty. It would serve as a warning to users in a way that text in the manuscript would not because it would travel with the timescale data file

- Add a section and a figure which specifically discusses the uncertainty and how it changes through the core.

*The reviewer is correct that the uncertainties in the output age model are too low. As discussed we have raised the uncertainty on some of the tie points. However the key issue is what happens between tie points where the age-depth relationship may be highly non-linear. To allow for this we have, in our new age model, raised the uncertainty on the thinning function. This of course cannot allow for the possibility of age reversals. However we have enough clear tie points in the section 117-126 ka to rule out any large scale reversals in this section. The change that will be made in response to this comment is to increase the uncertainties in tie points and thinning model, leading to little change in the age model itself but to larger and more realistic uncertainties that will be shown in the age model file. We also plan to add, in Fig. 13, a panel showing the age uncertainty.*

**Suggestions for the presentation of the LIG timescale:**

The figures and discussion of the timescale older than 106 ka can be revised to make it even easier to follow. Many of these figures should be integrated with subpanels because trying to find and compare the figures was challenging. At one point, I had a printed version open to three different pages and two electronic versions open on my screen. Here's a couple of suggestions:

- Provide a plot of the methane and d18O measurements below 600m by depth. The methane variations are not visible in Figure 3 because it spans the full core. When the methane is already matched and put on the timescale (e.g. Figure 5), the record can be deceiving.

- The methane-d18O cross plots should be combined into fewer figures. I suggest combining Figures 8 and 9 so that is much easier to compare the reference records with the cross plot. The cross-plot should also be extended to start at ~60 ka, such that the two intervals at ~80ka and ~100 ka, which have similar d18O-O2 and ch4 values similar to the LIG, are shown. Figures 10 and 11 should be combined with subpanels and have the same length of time shown – i.e. both run from 100 to 130 ka so that the colors remain the same in the two cross plots.

*We apologise if we made the relationship of the figures to the text too complicated. We have tested a number of ways of combining panels to see what might work best. We did also test a number of options for the time period covered in each cross-plot. If the age range is too large it becomes impossible to discern the age across the interglacial, so we prefer to keep it narrow but provide the context of the longer timescale as shown in what was Fig.9. Eventually what we have decided to do is as follows:*
*a) We have combined Figs 8 and 9 into one figure as they provide different visualisations of the same data (this was proposed by Rev. 2).*
*b) We have extended what was Fig. 9 from 0-200 ka to give the full context, and we have also plotted onto it the points we assign with definite LIG ages, so that the reader can see how unusual the values are in the long context.*
*c) We have also added a version of this figure with a different colour palette in the supplementary data. This figure is particularly hard to interpret because there is such a lot of data on it, so we believe that having the full range of colours used in the main text is justified but we acknowledge that the alternative may be easier to interpret for some readers (including those with colour vision issues).*
*d) We have also combined what was Figs. 10 and 11, using the same time range (100-130 ka) for both. We kept what was Fig. 12 separate because this necessarily must show a longer time range and so there is no advantage to combining it.*
*e)As requested we have added a new Supplementary Figure 1 which shows methane and 18O_atm data on a depth scale below 600 m.*

*We hope these changes do meet the request of the reviewer in a satisfying way. We have pasted the two new combined figures below.*

**Other comments:**

L503 – Can you rule out ice from the Ellsworth mountains flowing to Skytrain ice rise at a time when Skytrain was lower in elevation (and not an ice rise)? Are the elevations Ellsworths and its glaciers not sufficient to flow onto the bedrock beneath Skytrain? Or are the water isotope values in the Ellsworth mountains too cold? You have demonstrated that there is a stratigraphic disturbance, but I think stating that the "only plausible" explanation is due to contrasting rheology is too strong – particularly since there are no ice fabric measurements presented in the manuscript.

*Our main point was that it seems impossible that ice from inland WAIS could have come over the Ellsworths (with peaks of around 2000 m asl behind Skytrain) to reach Skytrain. As the reviewer hints, if it did it would have had to originate at very high altitude, with highly negative water isotope ratios which we do not observe. As we stated the bedrock of Skytrain is a high point a few hundred metres above sea level surrounded by a very deep bathymetry (1000 m in the two inlets and several hundred metres in the inland section in between). This makes it impossible for valley glaciers originating in the Ellsworth Mountains to reach*

*across to Skytrain. However we have not carried out ice sheet modelling to prove these assertions, and agree that "only plausible" is too strong a wording and we propose changing it to "most plausible". We also propose to add a sentence "However, detailed ice sheet modelling, as well as rheological studies on the Skytrain ice core, are required to firmly rule out other causes."*

*Revised figures are shown below (captions and figure numbering will of course be changed to match):*

---

## Author Comment (AC2)

Review in normal type, response in *italic*:

Review of ***The ST22 chronology for the Skytrain Ice Rise ice core – part 2: an age model to the last interglacial and disturbed deep stratigraphy***

Authors: Robert Mulvaney, Eric W. Wolff, et al

General Comments: The manuscript presents the age-scale for the deep section of the Skytrain Ice Rise, below the depths where annual layer counts are evident. The methodology chosen by the authors is robust and is similar to that chosen for other Antarctic ice cores. The manuscript is well written with generally good presentation. Relevant work is properly cited. I recommend publishing pending minor changes.

*We thank the reviewer for this overall positive assessment.*

- I am not sure of the claim that Skytrain is the only WAIS ice core that extends into MIS5 (the last interglacial period). Siple Dome extends to at least 100 ka (Severinghaus et al, 2009) and maybe to 118 Ka (Dunbar et al, 2011). RICE likely contains ice from ~90 ka. Taylor Glacier, which is on the border between East and West Antarctic similar to Hercules Dome, has been measured for MIS5-4 transition (Menking et al, 2019).

*This comment arises from a difference in terminology. The LIG is normally identified as MIS5e, ~110-130 ka, and not as the whole of MIS5 – we make this identification in line 43 of the paper. However we had missed that for Siple Dome some data extended to 100 ka, and we have now corrected our comment on this core and added the Severinghaus reference.*

- Beyond establishing the ST22 age scale, the authors hypothesize that ice from the last interglacial period is inherently prone to stratigraphic disturbances (e.g. folding). The Skytrain ice core is poorly suited for testing this hypothesis since this age range is within the bottom 50 m of the core, near the basal interface and stratigraphic disturbances may be expected. The authors do not convincingly demonstrate that the discontinuities observed in the Skytrain chronology are due to processes related to the age of the ice.

*We did not mean to sound so definite; the reviewer is correct that we do not have enough evidence to demonstrate the cause of stratigraphic disturbance convincingly. We propose to add a new sentence "We do not have enough evidence to conclude whether the disturbance we see is indeed due to rheogical contrasts or is just a consequence of investigating ice that is close to the bed." In response to the other reviewer we also changed the final sentence of section 9 from "only plausible" to "most plausible" in describing the cause of our disturbances.*

- Several of the figures could be combined into panels such as figures 4-7, all of which show how tie points were matched using reference records, and figures 9-12, which all show CH4 and d18O cross plots.

*As also requested by the other reviewer we will combine Figs 8 and 9 into one, and also Figs 10 and 11. We explain in our reply to Rev. 1 exactly the changes made in the figures, as well as our reasoning. We do not think it is very helpful to combine figures 4-7 as they refer to different time periods and follow the sections of the paper sequentially.*

Specific Comments:

22    "… Skytrain ice rise ice core." There should be some description of where this ice core is (Greenland vs Antarctica, Ross Sea Basin or Atlantic side of WAIS…)

*Done. Proposed text "We present an age model for the 651 m deep ice core from Skytrain Ice Rise ice core, situated inland of the Ronne Ice Shelf, Antarctica."*

32    "base of LIG section" is confusing. Please clarify.

*We are not really sure what the confusion is here. However we have added "below 628 m" after "LIG section" to avoid any doubt.*

38    "Antarctic Ice Sheet"  "Ice sheets of Antarctica". "Antarctic Ice Sheet" is a misnomer.

*We respectfully disagree. "Antarctic Ice Sheet" is a common and well-understood terminology for the whole ice sheet even if we often refer to component parts such as WAIS. Authorities such as the British Antarctic Survey (https://www.bas.ac.uk/about/antarctica/geography/ice/), National Geographic (https://education.nationalgeographic.org/resource/ice-sheet) and NSIDC (https://nsidc.org/learn/parts-cryosphere/ice-sheets) happily use the term. No change made.*

56-57  "The Siple Dome core reached the bed at 1004 m, but again data have only been presented as far back as 90 ka (Brook et al., 2005; Saltzman et al., 2006)" Severinghaus et al (2009) presents d18Oatm data back to 100 ka (973 m)

*Thank you for pointing this out. Corrected as above.*

58    In Lee et al (2020), there appear to have MIS5 ice near the base as identified by negative d18Oatm that are unique to interglacial periods.

*We very much hope that it will be possible to interpret deeper and older ice at RICE. However Lee et al does not provide us with a basis to claim that LIG ice (110-130 ka) is present.*

87    "(J. Kingslake, pers. Comm.)" Personal communication is not an acceptable reference. (https://www.climate-of-the-past.net/submission.html#references)

*I'm a little surprised at this interpretation of the rules. A pers comm is indeed not a reference, but it is a standard way to acknowledge the provider of information that is used in the paper but that does not warrant an authorship. However to avoid further argument, we will remove the pers comm from the text and instead acknowledge Kingslake in the acknowledgments.*

123    "as has been observed in other ice core records including those of the LIG in Greenland". See general comments.

*We have adjusted the text in section 9 as discussed above to cover the point made here by the reviewer.*

145    CFA has not yet been defined.

*Now defined here.*

154    "discrete samples analysed in Bern"   add something like "discussed below" or a reference for the discrete sample system.

*This sentence already refers to section 4.2 where the method in Bern is discussed.*

175    "ginput" MATLAB function. This function returns the [x,y] coordinates of a point plotted on a graph. It does not describe a method of identifying and removing outlier data. Do you mean, "manual identification"?

*Yes, we will alter to say that.*

Fig 3   It is hard to tell how well the CFA and discrete data represent each other. You could try a cross-plot as presented in Chappellaz et al (2013) Figure 6 and 7. Is it possible to "correct" the cfa data as they did for the NEEM record?

*The purpose of Fig 3 is to show that the CFA data are generally OK at the LIG, but are not OK between 534 and 545 m, as we suspected. Of course we could use the discrete data to offset the CFA data but this would assume that the offset stays constant and we don't feel we have enough discrete data to confirm that.  We therefore prefer to use the uncorrected CFA data as a pattern match rather than a value match, and then use the few discrete data to value match where they exist. We discuss the offset in section 4.1.  To satisfy the request to more easily compare the CFA and discrete methane, we now show a detail of Fig. 3 from 600-650 metres in a new Fig. S1.*

[Figure]

212    There should be a reference to a paper describing the system.

*The Extier paper referred to later in this paragraph describes the system for 18O_atm analysis.*

Fig 4    Why do the well known 5.2 and 8.2 ka methane variations not line up with the reference methane record?

*The 8.2 ka event lines up quite well in Fig 4, but we thank the reviewer for pointing out that there is a small mismatch. This arose from an incorrect entry of a tie point (the tie listed as 8226 should be at 8166). This will be corrected in the new version. A second incorrect tie point at 362.2 m also pulled the features around 6 ka slightly away from the reference data, and this is fixed in the new version. Table 3 and of course the age model are adjusted, and the figures affected will be replotted.*

Fig 5    Why are there such large age offsets between when nssMg is observed to increase in Skytrain at 90 and 105ka compared to EDML?

*We assume the reviewer means Fig 7 here. We did not use the nssMg in this region to create any tie points, because we did not want to place undue reliance on subtle features like this. The offset is therefore within reasonable uncertainty given that the nearest gas tie points are some way away and that delta-age is unconstrained.*

355-356       "…with some thousands of years potentially missing from our record" Are they missing or smoothed out? The reference record show d18Oatm approaching values of -0.4 permil at the peak of MIS5e. Based on the compressed age scale at this part of the core, a sample of 5-10 cm length could represent a significant amount of time. 5-10 cm is the length of typical discrete CH4 samples (Schmidley et al 2020) and 5-6 cm was typical system response of CFA CH4 (Rhodes 2017). This smoothing is on top of uncertainty of interpolation caused by measuring d18Oatm on adjacent samples, not on identical depths. Basically, I question whether uncertainty in age horizons are properly handled and suggest that the authors provide more discussion.

*The discrete samples we provided were generally 4 cm for 18O_atm adjacent to 3 cm for methane. In the lower dated section of the LIG 1 metre is about 1300 years so the combined samples for an 18O_atm/methane pair cover about a century.  The uncertainties we put on tie points using 18O_atm/methane are larger than this. However as discussed in the response to rev 1 we have increased the uncertainty on some of the methane ties around 70-100 ka where low data quality from the CFA mean that pinpointing the exact tie points in observed features is difficult, and our original picks were too tightly constrained.*

Figure 9       This should probably show the cross-plot for age range 60-140 ka as the authors need to prove that the ice is not just folded duplicates of MIS3, 4, 5a.

*What was Fig. 9 will now be provided as a pair with Fig 8, and covering the entire period 0-200 ka. We have included the Skytrain data points for the part we date as being at the main part of the LIG so the reader can see their position on the crossplot in the longer context.*

Figure 13      This would be nice to also show age uncertainty (as is output by ICECHRONO) and annual layer thickness as is provided in Buiron et al.  2011, Lee et al 2020). These are important metrics in understanding an ice core chronology.

*We plan to add, in Fig. 13, a panel showing the age uncertainty. We will also add a new supplementary figure with annual layer thickness.*

515 Were there d15N-N2 measurements made in parallel to the d18Oatm? Wouldn't this give you some idea of the accumulation rate?

*We do have some 15N data although the analysts of these samples were not very happy with the reproducibility of the 15N data. We assume the reviewer is asking if we have any very low values indicating a hiatus in accumulation. None of the values we have around the LIG are unusual in the context of the shallower ice.*

518 "bubble-free ice" could be confused with clathrated ice.

*Skytrain is not deep enough for clathrate formation to cause bubble-free ice. But the reviewer is of course correct that clathrated ice might be mistaken for bubble-free ice due to melt. Since our resut is negative (no bubble-free ice) though, this is not an issue for us.*

**Add'l References:**

Buiron, D., Chappellaz, J., Stenni, B., Frezzotti, M., Baumgartner, M., Capron, E., Landais, A., Lemieux-Dudon, B., Masson-Delmotte, V., Montagnat, M., Parrenin, F., and Schilt, A.: TALDICE-1 age scale of the Talos Dome deep ice core, East Antarctica, Clim. Past, 7, 1–16, https://doi.org/10.5194/cp-7-1-2011, 2011

Chappellaz, J., Stowasser, C., Blunier, T., Baslev-Clausen, D., Brook, E. J., Dallmayr, R., Faïn, X., Lee, J. E., Mitchell, L. E., Pascual, O., Romanini, D., Rosen, J., and Schüpbach, S.: High-resolution glacial and deglacial record of atmospheric methane by continuous-flow and laser spectrometer analysis along the NEEM ice core, Clim. Past, 9, 2579–2593, https://doi.org/10.5194/cp-9-2579-2013, 2013.

Dunbar, Nelia W., and Andrei V. Kurbatov. "Tephrochronology of the Siple Dome ice core, West Antarctica: correlations and sources." Quaternary Science Reviews 30.13-14 (2011): 1602-1614.

Menking, J. A., Brook, E. J., Shackleton, S. A., Severinghaus, J. P., Dyonisius, M. N., Petrenko, V., McConnell, J. R., Rhodes, R. H., Bauska, T. K., Baggenstos, D., Marcott, S., and Barker, S.: Spatial pattern of accumulation at Taylor Dome during Marine Isotope Stage 4: stratigraphic constraints from Taylor Glacier, Clim. Past, 15, 1537–1556, https://doi.org/10.5194/cp-15-1537-2019, 2019.

Severinghaus, Jeffrey P., et al. "Oxygen-18 of O2 records the impact of abrupt climate change on the terrestrial biosphere." Science 324.5933 (2009): 1431-1434.

---

## Author Response (AR2)

**Author response, Mulvaney, Wolff et al**

There were two requests, one from the editor and one from the editorial team.

In response to the editor, we have added the links (including doi) for the Pangaea files in the data availability statement.

In response to the request about colour of figures, we checked all figures. Although all were understandable under different vision impairment conditions, we nonetheless changed the colour scheme in Figs 3, 4 and 5, to ensure greater discernment between lines. For Figs 8b, 9 and 10, we have supplied versions with a different colour scheme in the supplement in addition to those in the main text. The ones in the supplement are accessible to those with all types of colour vision impairment. However given the complexity of the figures we feel there is value in making available figures that those without impairment will more easily appreciate in the main text. We hope this is acceptable.

Section numbers were redone thoughout to avoid listing the abstract as section 1 (a mistake in the last version).